# Exhaustive Photocatalytic Lindane Degradation by Combined Simulated Solar Light-Activated Nanocrystalline TiO₂ and Inorganic Oxidants

**Sanaullah Khan [1,2,3,†], Changseok Han [4,†] , Murtaza Sayed [2] , Mohammad Sohail [5], Safeer Jan [2,6] , Sabiha Sultana [7], Hasan M. Khan [2] and Dionysios D. Dionysiou [3,8,*]**

1   Department of Chemistry, Women University Swabi, Swabi 23430, Pakistan; sukhan3@gmail.com
2   Radiation and Environmental Chemistry Laboratory, National Centre of Excellence in Physical Chemistry, University of Peshawar, Peshawar 25120, Pakistan; murtazasayed_407@yahoo.com (M.S.); safeerchemist@yahoo.com (S.J.); hmkhan3@gmail.com (H.M.K.)
3   Environmental Engineering and Science Program, Department of Chemical and Environmental Engineering (ChEE), University of Cincinnati, Cincinnati, OH 45221-0012, USA
4   Department of Environmental Engineering, INHA University, Incheon 22212, Korea; hanck@inha.ac.kr
5   Institute of Chemical Sciences, University of Swat, Swat 19130, Pakistan; msohail2000@gmail.com
6   Hubei Key Laboratory of Electrochemical Power Sources, College of Chemistry and Molecular Sciences, Wuhan University, Wuhan 430072, China
7   Department of Chemistry, Islamia College University Peshawar, Peshawar 25120, Pakistan; saba_rehan2006@yahoo.com
8   Nireas-International Water Research Centre, University of Cyprus, 1678 Nicosia, Cyprus
*   Correspondence: dionysios.d.dionysiou@uc.edu; Tel.: +1-513-556-0724; Fax: +1-513-556-2599
†   Contributed equally to this work.

**Abstract:** Organochlorine compounds (OCs) are very toxic, highly persistent, and ubiquitous contaminants in the environment. Degradation of lindane, a selected OC, by simulated solar light-activated TiO₂ (SSLA-TiO₂) photocatalysis was investigated. The film types of the TiO₂ photocatalyst were prepared using a dip-coating method. The physical properties of the films were investigated using X-ray diffraction, transmission electron microscopy, and environmental scanning electron microscopy. The SSLA-TiO₂ photocatalysis led to a lindane removal of 23% in 6 h, with 0.042 h⁻¹ of an observed pseudo first-order rate constant ($k_{obs}$). The SSLA-TiO₂ photocatalysis efficiency was greatly enhanced by adding hydrogen peroxide (H₂O₂), persulfate (S₂O₈²⁻), or both combined, corresponding to a 64%, 89%, and 99% lindane removal in the presence of 200 μM of H₂O₂, S₂O₈²⁻, or equimolar H₂O₂-S₂O₈²⁻, respectively. The hydroxyl and sulfate radicals mainly participated in lindane degradation, proven by the results of a radical scavenger study. The degradation kinetics were hindered in the presence of the water constituents, indicated by a 61%, 35%, 50%, 70%, 88%, and 91% degradation of lindane in 6 h, using a SSLA-TiO₂/S₂O₈²⁻/H₂O₂ photocatalysis system containing 1.0 mg L⁻¹ humic acid (HA), or 1 mM of CO₃²⁻, HCO₃⁻, NO₃⁻, SO₄²⁻, and Cl⁻, respectively. The TiO₂ film demonstrated high reusability during four runs of lindane decomposition experiments. The SSLA-TiO₂/S₂O₈²⁻/H₂O₂ photocatalysis is very effective for the elimination of a persistent OC, lindane, from a water environment.

**Keywords:** lindane; simulated solar light; TiO₂ photocatalysis; S₂O₈²⁻/H₂O₂; natural water constituents; water treatment

---



## 1. Introduction

Many organochlorine compounds (OCs), such as chlorinated alkanes, alkenes, benzenes, phenols, and biphenyls, are often introduced into the environment in the form of solvents, disinfectants, soil fumigants, pesticides, and as dye precursors [1]. A considerable number of OCs also enter the environment as by-products from waste incineration, the chlorination of drinking water and wastewater, and bleaching of pulp with chlorine [2]. OCs are generally considered as very toxic, highly bioaccumulative, and strongly resistant towards biodegradation [2].

Hexachlorocyclohexanes (HCHs) constitute one of the largest classes of chlorinated alkanes that have extensively been used as organochlorine pesticides during the last several decades [3]. The HCHs consist of eight isomers, namely β, γ, δ, θ, ε, η, and two α-enantiomers. The insecticidal property of HCHs is solely due to the γ-isomer, commonly known as lindane (i.e., 99% γ-HCH) [4]. The low cost and high efficiency of lindane led to its excessive usage in a wide application range, such as an insecticide and a seed treatment agent in the agricultural and forest industry, a vector control in public health, and as anti-mice and anti-lice agents for livestock and domestic purposes [5,6]. Because of the high chemical stability and mobility of lindane, it ubiquitously disperses in the environment, including biota via the food chain [7]. Owing to the presence of large number of chlorine atoms in the molecule, lindane is a highly toxic compound and is recognized as an endocrine disruptor in the environment [8]. The development of more efficient methods for removing lindane from water is of fundamental importance for environmental cleanup.

Advanced oxidation processes (AOPs) are promising alternatives to the conventional water and wastewater treatment processes, owing to their wide versatility and high efficiency [9]. Among AOPs, $TiO_2$ photocatalysis is considered as one of the most promising technologies because of the low cost, easy availability, and environmentally benign nature of $TiO_2$ [10]. The solar light-induced $TiO_2$ photocatalysis has recently gained much attention in water decontamination and disinfection, owing to the availability and sustainability of sunlight radiation [11,12]. The photocatalytic performance of the catalysts is improved significantly with an increased crystallinity, large surface area, and tailor-designed morphology [13–15]. The efficiency of solar light-activated $TiO_2$ photocatalytic processes is usually low, because only a small portion of the sunlight (i.e., 5%, namely UV radiation) is involved in the activation of $TiO_2$ (Reaction (1)) [11].

$$TiO_2 + hv \rightarrow h_{VB}^+ \text{ (valence band hole)} + e_{CB}^- \text{ (conduction band electron)} \tag{1}$$

Hydrogen peroxide ($H_2O_2$), persulfate ($S_2O_8^{2-}$), and, more recently, peroxymonosulfate ($HSO_5^-$) are emerging inorganic oxidants employed or explored in water and/or wastewater treatment processes, owing to the generation of the strongly reactive oxidants of the hydroxyl radical ($^\bullet OH$) and sulfate radical ($SO_4^{\bullet -}$) [16–18]. Several inorganic anions (e.g., $CO_3^{2-}$, $HCO_3^-$, $NO_3^-$, $SO_4^{2-}$, and $Cl^-$) and organic acids (e.g., humic acid (HA)) are frequently found in water sources, owing to the natural abundances and man-made activities [19]. Depending on the mode of the reaction, these inorganic and organic constituents in water may differently affect the removal efficiency of pollutants using different AOPs [20]. Only limited information is available about the effect of these constituents on the efficiency of combined photocatalytic–photochemical processes, so far.

In this study, the efficiency of a simulated solar light-activated nanocrystalline $TiO_2$ photocatalyst for decomposing a selected OC, lindane, in an aqueous solution was investigated. The synthesized $TiO_2$ photocatalyst films were characterized using X-ray diffraction (XRD), environmental scanning electron microscopy (ESEM), and transmission electron microscopy (TEM), for determining its surface morphology and structural properties. The effect of $H_2O_2$ and $S_2O_8^{2-}$ on the activity of the simulated solar light-activated $TiO_2$ (SSLA-$TiO_2$) photocatalysis for the removal of lindane was investigated. Radical scavenging experiments were conducted so as to examine the relative importance of the reactive species towards the degradation of lindane. The effect of natural water constituents (i.e., HA and inorganic ions) on the efficiency of the SSLA-$TiO_2$/$S_2O_8^{2-}$/$H_2O_2$ process was investigated, considering

practical applications. Finally, the performance sustainability of the synthesized $TiO_2$ photocatalyst was evaluated using four runs of lindane decomposition experiments. The obtained results could provide useful data on the application of SSLA-$TiO_2$ photocatalysis for removing persistent OCs, such as lindane, from the water environment.

## 2. Results and Discussion

### 2.1. Characteristics of Synthesized TiO$_2$ Films

Figure 1a shows the ESEM image for the film of the $TiO_2$ photocatalyst. As the film was dried under infrared illumination and then slowly heated up to 350 °C, no cracks in the $TiO_2$ film were observed at a magnification of 2000×. The peaks of the XRD analysis of the synthesized $TiO_2$ corresponding to the anatase phase of $TiO_2$ were observed (Figure 1b). This indicates that the anatase phase of $TiO_2$ dominated in the films with the synthesis method, based on the observed peaks at 2θ (degree) = 24.8, 37.3, 47.6, 53.5, 55.1, 62.2, 68.8, 70.1, and 74.9, corresponding to the (101), (004), (200), (105), (211), (204), (116), (220), and (215) planes, respectively, of anatase $TiO_2$ (JCPDS card no. 21-1272). Figure 1c,d shows the HR-TEM images of the sample. The average crystal size of the $TiO_2$ photocatalyst was 20 ± 5.7 nm. The measured BET surface area was 13.1 ± 0.04 $m^2$ $g^{-1}$ (Figure 1e,f). The spacing between the lattice fringes was calculated as 0.351 nm, similar to the $TiO_2$ anatase lattice spacing of the (101) plane (i.e., 0.352 nm) [21]. The calculated band-gap energy of the synthesized $TiO_2$ film by the Kubelka–Munk transformation was 3.1 eV (Figure S1). These results confirmed the formation of the anatase crystalline phase, which has a better photocatalytic activity compared to rutile [22]. The small particle size with the corresponding high surface area, as well as the large number of active sites, could lead to the high photocatalytic activity of the $TiO_2$ anatase crystalline phase [23]. Lin et al. [24] also reported that the band gap of the $TiO_2$ nanoparticles is a function of primary particle size, so that when the $TiO_2$ particle size decreased (i.e., 29 to 17nm), the band gap decreased as well.

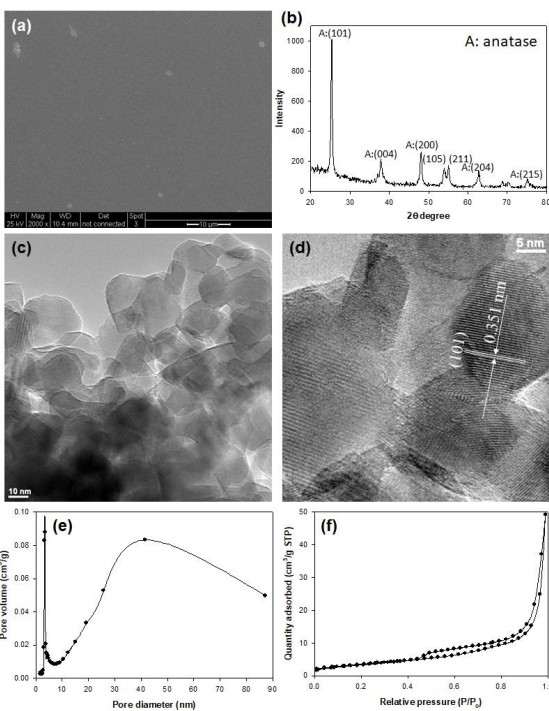

**Figure 1.** (**a**) Environmental scanning electron microscopy (ESEM) image of the $TiO_2$ film, (**b**) X-ray diffraction (XRD) spectrum of the $TiO_2$ film, (**c,d**) high-resolution transmission electron microscopy (HR-TEM) images of the $TiO_2$ film, (**e**) pore size distribution, and (**f**) $N_2$ adsorption–desorption isotherm of $TiO_2$ film.

The valence band electrons of the synthesized $TiO_2$ film are excited to a conduction band when a solar light of sufficient energy illuminates the film. The electrons in the conduction band ($e_{CB}^-$) interact with oxygen to generate the highly reducing superoxide radical anion ($O_2^{\bullet-}$) (i.e., Reaction (2)), while the valence band holes ($h_{VB}^+$) react with $H_2O$ or $OH^-$ producing the strongly oxidizing $^\bullet OH$, following Reactions (3) and (4), respectively [10].

$$e_{CB}^- + O_2 \rightarrow O_2^{\bullet-} \tag{2}$$

$$h_{VB}^+ + H_2O \rightarrow {}^\bullet OH + H^+ \tag{3}$$

$$h_{VB}^+ + OH^- \rightarrow {}^\bullet OH \tag{4}$$

### 2.2. SSLA-TiO₂ Photocatalysis of Lindane

Figure 2 shows the degradation of lindane by SSLA-TiO$_2$ photocatalysis, indicating that 23% lindane was removed in 6 h, corresponding to an observed rate constant ($k_{obs}$) of 0.042 h$^{-1}$. Many papers were published on the SSLA-TiO$_2$ photocatalysis of organic compounds [25–27].

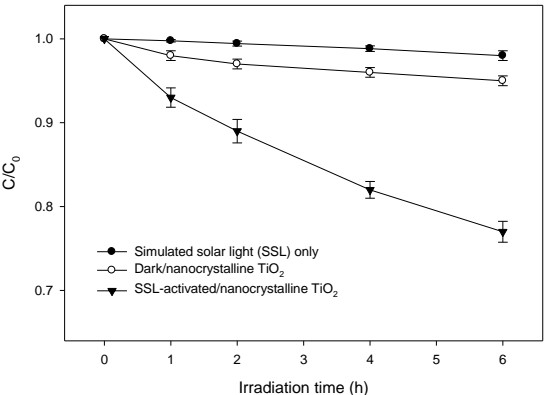

**Figure 2.** Simulated solar light-activated TiO$_2$ photocatalysis of lindane. [lindane]$_0$ = 1.0 µM; mass of TiO$_2$ film = 9.02 mg; thickness of TiO$_2$ film = 1.02 µm; area of TiO$_2$ film = 3750 mm$^2$; pH = 5.8.

The $^\bullet OH$ produced in Reactions (3) and (4) was mainly responsible for the decomposition of organic contaminants [28,29]. The $O_2^{\bullet-}$ formed in Reaction (2) might also participate in pollutant degradation, and may likely react with a water molecule to produce additional $^\bullet OH$ [30]. Khan et al. [31] previously reported an efficient lindane degradation by $^\bullet OH$, besides a minor lindane removal by $O_2^{\bullet-}$, employing simulated solar light-activated sulfur-doped TiO$_2$ (S-TiO$_2$) photocatalysis. In the current case too, it was seen that $^\bullet OH$ mainly participated in lindane degradation, although $O_2^{\bullet-}$ also contributed slightly in the degradation process, as will be discussed latter in Section 2.4. Literature studies show that the degradation efficiency of some other organic compounds by solar TiO$_2$ photocatalysis was rather high [32–34]. The reaction rate constants for the SSLA-TiO$_2$ photocatalysis of some other pesticides, such as λ-cyhalothrin, chlorpyrifos, and diazinon (C$_0$ = 0.1, 0.57, and 0.36 mM, respectively) were 0.48, 0.44, and 0.25 h$^{-1}$, respectively [25]. Adishkumar and Kanmani [26] found that the reaction rate constant for SSLA-TiO$_2$ photocatalysis of phenol (C$_0$ = 1.06 mM) was 1.76 h$^{-1}$. The apparent discrepancy in the degradation rate constants could be because of the effect of the light intensity [35–37], as well as the molecular structure differences, as previously reported regarding organic pollutants degradation using other AOPs [38]. Parra et al. [36] showed that the degradation efficiency of atrazine using solar simulator/TiO$_2$ photocatalysis was significantly enhanced by increasing the solar light intensity from 50 to 90 mW/cm$^2$. Khataee and Kasiri [39] reported that monoazo dyes have higher photocatalytic degradation rates as compared to the dyes with an antraquinone structure. The presence of a methyl or chloro group in the dye molecule showed a decreasing effect, while the nitrite group

has an increasing effect on the degradation efficiency. The photocatalytic efficiency of dye molecules decreases by a sulfonic substituent, while the hydroxyl group has an opposite effect [39].

Zaleska et al. [40] reported a 77% lindane degradation ($[lindane]_0 = 0.137$ mM; $[TiO_2]_0 = 0.5$ g/L) after 2.5 h irradiation in the anatase $TiO_2$/UV photocatalytic process. The discrepancy between our results and those reported by Zaleska et al. [40] is in agreement with the findings by Parra et al. [36], showing a higher degradation efficiency of atrazine using UV/$TiO_2$ than solar simulator/$TiO_2$ photocatalysis, obviously due to an efficient activation of $TiO_2$ for $^{\bullet}$OH radical generation by UV than simulated solar light.

Despite the significant lindane degradation achieved in 6 h, the efficiency of the SSLA-$TiO_2$ process in the current case may be regarded as low, particularly considering the practical applications. In an attempt to achieve a higher degradation efficiency by SSLA-$TiO_2$ photocatalysis, inorganic oxidants, such as $S_2O_8^{2-}$ and $H_2O_2$, were employed as additives in the subsequent experiments, as discussed below.

## 2.3. Effect of $S_2O_8^{2-}$ and $H_2O_2$ on Lindane Degradation by SSLA-$TiO_2$ Photocatalysis

Figure 3 shows the effect of $S_2O_8^{2-}$ and $H_2O_2$ on the removal efficiency of lindane using SSLA-$TiO_2$ photocatalysis. The removal efficiency of lindane was remarkably enhanced by adding 200 μM of $S_2O_8^{2-}$ or $H_2O_2$, indicated by an 89% and 64% lindane removal in 6 h, corresponding to an observed pseudo first-order rate constant ($k_{obs}$) of 0.369 and 0.098 $h^{-1}$, respectively. Both $H_2O_2$ and $S_2O_8^{2-}$ are strong electron acceptors, capable of scavenging the photogenerated $e_{CB}^-$, according to Reactions (5) and (6), respectively, thus increasing the concentration of $h_{VB}^+$, with a subsequent higher $^{\bullet}$OH concentration on the $TiO_2$ surface [41]. The photolysis of $H_2O_2$ and $S_2O_8^{2-}$ can lead to the generation of additional $^{\bullet}$OH, as well as $SO_4^{\bullet-}$, according to Reactions (7) and (8), respectively [42,43]. The above-mentioned phenomena, that is, the promotion of the charge separation followed by an increased production of $^{\bullet}$OH and the generation of additional $^{\bullet}$OH and $SO_4^{\bullet-}$, could explain the enhancing effect of $H_2O_2$ and $S_2O_8^{2-}$ [44]. The comparatively higher enhancing effects exerted by the $S_2O_8^{2-}$ than $H_2O_2$ were probably due to an easier activation of $S_2O_8^{2-}$ than $H_2O_2$ by simulated solar radiation containing UV light [45]. The quantum yield results show that the concentration of $SO_4^{\bullet-}$ generated by the simulated solar radiation (containing UV light) activation of $S_2O_8^{2-}$ was high, compared to that of $^{\bullet}$OH that resulted from $H_2O_2$ [46]. Antonopoulou and Konstantinou [47], and Koltsakidou et al. [32] also reported that the addition of $S_2O_8^{2-}$ showed a stronger enhancing effect than $H_2O_2$ on the decomposition of N,N-diethyl-m-toluamide and cytarabine, respectively, by simulated solar-light assisted $TiO_2$ photocatalysis.

$$H_2O_2 + e_{CB}^- \rightarrow {}^{\bullet}OH + HO^- \tag{5}$$

$$S_2O_8^{2-} + e_{CB}^- \rightarrow SO_4^{\bullet-} + SO_4^{2-} \tag{6}$$

$$H_2O_2 + h\nu \rightarrow 2\,{}^{\bullet}OH \ (\lambda = 253.7 \text{ nm}, \phi = 1.0) \tag{7}$$

where "$\phi$" is the quantum yield, that is, the number of species formed per photon absorbed.

$$S_2O_8^{2-} + h\nu \rightarrow 2SO_4^{\bullet-} \ (\lambda = 253.7 \text{ nm}, \phi = 1.8) \tag{8}$$

$${}^{\bullet}OH + {}^{\bullet}OH \rightarrow H_2O_2 \ (k = 5.3 \times 10^9 \text{ M}^{-1} \text{ s}^{-1}) \tag{9}$$

$$SO_4^{\bullet-} + SO_4^{\bullet-} \rightarrow S_2O_8^{2-} \ (k = 4 \times 10^8 \text{ M}^{-1} \text{ s}^{-1}) \tag{10}$$

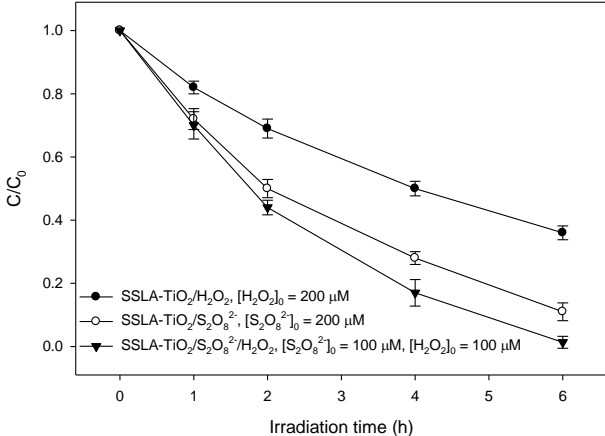

**Figure 3.** Effect of $S_2O_8^{2-}$ and $H_2O_2$ additives on the simulated solar light-activated $TiO_2$ photocatalysis of lindane. $[lindane]_0 = 1.0$ µM; $[S_2O_8^{2-}]_0 = [H_2O_2]_0 = 200$ µM; mass of $TiO_2$ film = 9.02 mg; thickness of $TiO_2$ film = 1.02 µm; area of $TiO_2$ film = 3750 mm$^2$; pH = 5.8.

Furthermore, an even stronger enhancing effect on the lindane degradation efficiency (Figure 3) was noticed, indicated by a 99% lindane removal in 6 h, corresponding to a $k_{obs}$ of 0.550 h$^{-1}$ when each 100 µM of $S_2O_8^{2-}$ and $H_2O_2$ was added into the system together. Table 1 shows the kinetics analyses of degradation of various pesticides using $TiO_2$ photocatalytic processes under different conditions [25,48].

**Table 1.** Kinetics analyses of degradation of pesticides using different photocatalytic processes.

| Pesticides | $k_{obs}$ (h$^{-1}$) | Reaction Type | Reference |
|---|---|---|---|
| Vinclozoline | 0.408 | a | [48] |
| Quinalphos | 0.426 | a | [48] |
| Malathion | 4.266 | a | [48] |
| Fenarimol | 0.168 | a | [48] |
| Fenitrothion | 0.300 | a | [48] |
| Dimethoate | 0.666 | a | [48] |
| Lambda-Cyhalothrin | 0.504 | b | [25] |
| Chlorpyrifos | 0.498 | b | [25] |
| Diazinon | 0.270 | b | [25] |
| Lindane | 0.550 | c | this study |

[a] Natural sunlight/$TiO_2$/$S_2O_8^{2-}$; $[S_2O_8^{2-}]_0 = 250$ mg/L; $TiO_2$ (P25) = 200 mg/L. [b] Natural sunlight/$TiO_2$/$H_2O_2$; $[H_2O_2]_0 = 1000$ mg/L; $TiO_2 = 2.0$ g/L. [c] Simulated solar light/$TiO_2$/$S_2O_8^{2-}$/$H_2O_2$; $[S_2O_8^{2-}]_0 = [H_2O_2]_0 = 100$ µM. Mass of $TiO_2$ film = 9.02 mg; thickness of $TiO_2$ film = 1.02 µm; area of $TiO_2$ film = 3750 mm$^2$.

The degradation of organic compounds usually takes place in several steps, involving the generation and destruction of various reaction by-products [49]. The various reaction by-products of lindane identified in the SSLA-$TiO_2$/$S_2O_8^{2-}$/$H_2O_2$ process included hexachlorobenzene, tetrachlorocyclohexene, and dichlorophenol, probably resulting via hydrogen abstraction and hydroxylation pathways by using $SO_4^{\bullet-}$ and $^{\bullet}OH$ [20,31]. All of the generated by-products eventually disappeared after 6 h of irradiation. The reactivity of the various by-products generated during the degradation process could be different towards $^{\bullet}OH$ and $SO_4^{\bullet-}$ [50]. For example, chlorobenzene, which is identified in this study as well as frequently reported as a lindane by-product elsewhere using $TiO_2$ photocatalysis [31,51], typically showed a higher rate constant towards $^{\bullet}OH$ than $SO_4^{\bullet-}$, probably because of the high tendency of the former species for addition reactions due to the multiple bonds [50]. Literature studies show that the efficiency of UV/$S_2O_8^{2-}$ and UV/$H_2O_2$ systems that are capable of generating $SO_4^{\bullet-}$ and $^{\bullet}OH$, respectively, varied when applied to different types of chemical compounds [52,53]. This might explain the larger enhancing effect while using both $S_2O_8^{2-}$ and $H_2O_2$

combined, because it could provide an opportunity for the different by-products generated to be efficiently degraded under the conditions favorable to it, that is, either $SO_4^{\bullet-}$ or $^{\bullet}OH$, or both combined.

Furthermore, the residual oxidant analysis revealed that out of the 100 μM of $H_2O_2$ and $S_2O_8^{2-}$ each employed in the $SSLA-TiO_2/S_2O_8^{2-}/H_2O_2$ process, 7 and 23 μM were left as a residue after 6 h of treatment, respectively.

Figure 4 shows the effect of the initial concentration of $S_2O_8^{2-}$ and $H_2O_2$ on the observed pseudo first-order rate constant ($k_{obs}$) of lindane using the $SSLA-TiO_2$ photocatalysis. The value of $k_{obs}$ increased at a higher initial $S_2O_8^{2-}$ and $H_2O_2$ concentration, attributable to an increased promotion of the charge separation, as well as the higher concentrations of $SO_4^{\bullet-}$ and $^{\bullet}OH$. However, the increase in $k_{obs}$ was less in the case of $H_2O_2$ than $S_2O_8^{2-}$, probably due to the increased recombination rate in the former case, that is, the higher second-order rate constant of Reaction (9) than Reaction (10) [20]. Bekkouche et al. [54] determined an enhancing effect due to an increasing initial concentration of $S_2O_8^{2-}$ on the solar-$UV/TiO_2/S_2O_8^{2-}$ photocatalytic degradation of Safranin O, attributed to an increased generation of reactive radicals. Saien et al. [45] reported the effects of the initial concentration of $S_2O_8^{2-}$ and $H_2O_2$ using the $UV/TiO_2$ photocatalysis of Triton X-100, showing an enhancing effect as a result of increasing the initial concentrations of the oxidants. Koltsakidou et al. [32] reported a decreasing effect as a result of increasing the initial concentrations of $H_2O_2$ on the $SSLA-TiO_2$ photocatalytic degradation of cytarabine, attributed to the fast scavenging of $^{\bullet}OH$ at comparatively higher $H_2O_2$ concentrations (i.e., 1–4 mM).

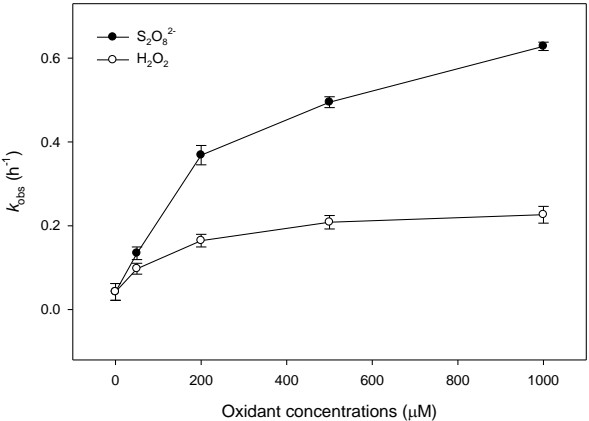

**Figure 4.** Variation of $k_{obs}$ with increasing initial concentration of $S_2O_8^{2-}$ and $H_2O_2$ on the simulated solar light-activated $TiO_2$ photocatalysis of lindane. $[lindane]_0$ = 1.0 μM; mass of $TiO_2$ film = 9.02 mg; thickness of $TiO_2$ film = 1.02 μm; area of $TiO_2$ film = 3750 mm$^2$; pH = 5.8.

## 2.4. Radical Scavenger Studies and Role of the Reactive Species

The $^{\bullet}OH$, $SO_4^{\bullet-}$, and $O_2^{\bullet-}$ are the main reactive species present in the $SSLA-TiO_2/S_2O_8^{2-}/H_2O_2$ system, as discussed above. To identify the role of the individual reactive species in the lindane decomposition, appropriate radical scavenger experiments were performed.

Benzoquinone, tert-butanol, and iso-propanol were employed to scavenge $O_2^{\bullet-}$, $^{\bullet}OH$, and both $^{\bullet}OH$ and $SO_4^{\bullet-}$, respectively, according to Reactions (11)–(14) [50,55]. In the presence of 50 mM benzoquinone, 89% lindane was decomposed by $SSLA-TiO_2/S_2O_8^{2-}/H_2O_2$ in 6 h (Figure 5). In the absence of a radical scavenger, a 99% removal of lindane could be achieved by $SSLA-TiO_2/S_2O_8^{2-}/H_2O_2$ in 6 h. The result showed a 10% decrease in the lindane removal efficiency as a result of the addition of benzoquinone, attributable to the reactions of $O_2^{\bullet-}$. By adding 50 mM tert-butanol, a 58% removal of lindane was achieved in 6 h (Figure 5), indicating a 41% loss in the efficiency of the $SSLA-TiO_2/S_2O_8^{2-}/H_2O_2$ system, attributable to the role of $^{\bullet}OH$. In the presence of 50 mM iso-propanol, only an 11% removal of lindane occurred in 6 h (Figure 5), indicating an 88% decrease in the removal efficiency of lindane by $SSLA-TiO_2/S_2O_8^{2-}/H_2O_2$, attributable to the role of the combination of $^{\bullet}OH$ and $SO_4^{\bullet-}$. By subtracting, the role of $^{\bullet}OH$ and $SO_4^{\bullet-}$ was found to be 41% and 47%, respectively.

The relatively larger contribution of $SO_4^{\bullet-}$ than $^{\bullet}OH$ could be due to the high concentration of the former species [45], as well as its high rate constant with lindane [20].

$$\text{Benzoquinone} + O_2^{\bullet-} \rightarrow \text{Benzoquione}^{\bullet-} + O_2 \quad k = 9.6 \times 10^8 \text{ M}^{-1}\text{ s}^{-1} \tag{11}$$

$$(CH_3)_3COH + {}^{\bullet}OH \rightarrow (CH_3)_2{}^{\bullet}CH_2COH + H_2O \quad k = 5.2 \times 10^8 \text{ M}^{-1}\text{ s}^{-1} \tag{12}$$

$$(CH_3)_2CHOH + {}^{\bullet}OH \rightarrow (CH_3)_2{}^{\bullet}COH + H_2O \quad k = 1.9 \times 10^9 \text{ M}^{-1}\text{ s}^{-1} \tag{13}$$

$$(CH_3)_2CHOH + SO_4^{\bullet-} \rightarrow (CH_3)_2{}^{\bullet}COH + HSO_4^{-} \quad k = 8.2 \times 10^7 \text{ M}^{-1}\text{ s}^{-1} \tag{14}$$

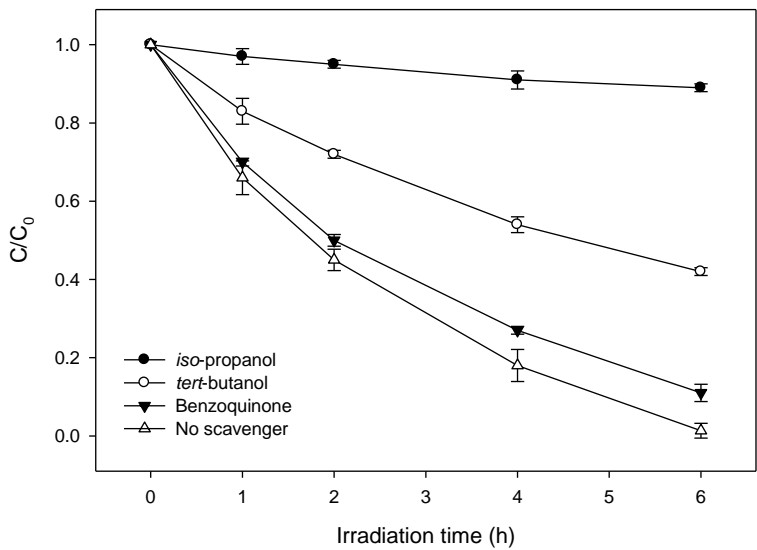

**Figure 5.** The role of $^{\bullet}OH$, $SO_4^{\bullet-}$, and $O_2^{\bullet-}$ in the simulated solar light-activated $TiO_2/S_2O_8^{2-}/H_2O_2$ photocatalysis of lindane. $[\text{lindane}]_0 = 1.0$ μM; $[\text{iso-propanol}]_0 = 50$ mM; $[\text{tert-butanol}]_0 = 50$ mM; $[\text{benzoquinone}]_0 = 50$ mM; mass of $TiO_2$ film = 9.02 mg; thickness of $TiO_2$ film = 1.02 μm; area of $TiO_2$ film = 3750 mm²; pH = 5.8.

## 2.5. Influence of Natural Water Constituents

Inorganic ions are one of the most frequently found ingredients in water and wastewater, and are derived from various anthropogenic and non-anthropogenic sources [19]. The presence of inorganic ions can influence the photocatalytic and/or photochemical degradation of organic compounds [56–59]. $CO_3^{2-}$, $HCO_3^{-}$, $NO_3^{-}$, $SO_4^{2-}$, and $Cl^{-}$, typical components of natural waters, were selected as inorganic ions in this study. As seen in Figure 6, the efficiency of the SSLA-$TiO_2/S_2O_8^{2-}/H_2O_2$ system was reduced by 65%, 50%, 30%, 12%, and 9% in the presence of 1.0 mM of $CO_3^{2-}$, $HCO_3^{-}$, $NO_3^{-}$, $SO_4^{2-}$, and $Cl^{-}$, respectively. The inorganic ions could be adsorbed onto the $TiO_2$ catalyst, thereby decreasing the active site number on its surface [60]. The inhibiting effect may be due to the reduced generation of the reactive radicals owing to the blocking of the $TiO_2$ active sites [61], as well as the scavenging of the reactive radicals by the ions [59,62]. Liang et al. [58] reported that a degradation rate of the $TiO_2$ photocatalysis of 2,3-dichlorophenol decreased in the presence of $NO_3^{-}$, $Cl^{-}$, $SO_4^{2-}$, and $HPO_4^{-}$, attributed to the competitive adsorption on the surface of $TiO_2$, besides the scavenging of the $^{\bullet}OH$ by the ions. Muruganandham and Swaminathan determined that the UV/$TiO_2$ photocatalytic oxidation of Reactive Yellow 14 was decreased by adding $Cl^{-}$ or $CO_3^{2-}$, attributed to the $^{\bullet}OH$ scavenging [63]. Our previous study [64] reported a 51%, 34%, 3%, and 1% decrease on the lindane removal efficiency by UV/$HSO_5^{-}$ process in the presence of 1.0 mM $CO_3^{2-}$, $HCO_3^{-}$, $Cl^{-}$, or $SO_4^{2-}$, respectively, which was attributed to the scavenging of $SO_4^{\bullet-}$ and $^{\bullet}OH$. Contrary to our previous study involving lindane degradation by an UV/$HSO_5^{-}$ system [64], the comparatively larger inhibiting effect observed in the current case may indicate an additional $TiO_2$ deactivation by the ions, besides the scavenging of the

reactive radicals, according to Reactions (15)–(19) [50,65,66]. Despite the high affinity of $Cl^-$ for the scavenging of reactive radicals (Reactions (18) and (19)), a rather small inhibiting effect was observed on the degradation efficiency of lindane. A plausible reason could be the involvement of the reactive $Cl^\bullet$ in the degradation process, as was previously described [20].

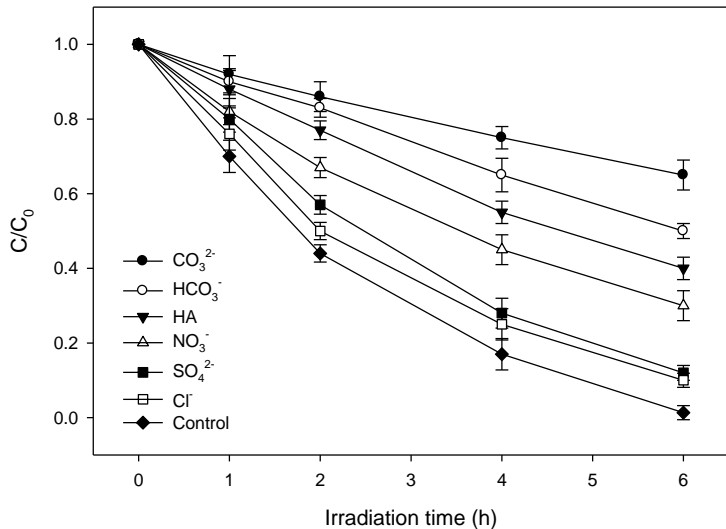

**Figure 6.** Effect of humic acid and inorganic anions ($CO_3^{2-}$, $HCO_3^-$, $NO_3^-$, $SO_4^{2-}$, and $Cl^-$) on the SSLA-$TiO_2$/$S_2O_8^{2-}$/$H_2O_2$ photocatalysis of lindane. [lindane]$_0$ = 1.0 μM; mass of $TiO_2$ film = 9.02 mg; thickness of $TiO_2$ film = 1.02 μm; area of $TiO_2$ film = 3750 mm$^2$; [humic acid]$_0$ = 1 mg/L; [inorganic ions]$_0$ = 1 mM.

The efficiency of lindane degradation by the SSLA-$TiO_2$/$S_2O_8^{2-}$/$H_2O_2$ process decreased by 39% in the presence of 1.0 mg $L^{-1}$ HA (Figure 6). The result was consistent with the findings of Bekkouche et al. [54], showing that the solar-UV/$TiO_2$/$S_2O_8^{2-}$ photocatalytic degradation of Safranin O decreased in the presence of HA. Plausible reasons for the inhibitory effect could be (i) the scavenging of $^\bullet OH$ and $SO_4^{\bullet-}$ by HA (Reactions (20) and (21)) [67,68], (ii) the blocking of active sites on the $TiO_2$ surface attributable to HA adsorption [69], and (iii) the absorption of photons by HA [59]. Doorslaer et al. [70] reported that the UV/$TiO_2$ photocatalytic degradation rate of moxifloxacin decreased by adding different types of dissolved organic matter (DOM), including HA and fulvic acid, attributed to the scavenging of reactive species (i.e., $^\bullet OH$) as well as the absorption of UV light by DOM. The results might suggest that the natural water constituents studied may adversely affect the efficiency of the $TiO_2$/oxidant-based AOPs applied for the decontamination of field waters, which might need further attention.

$$HCO_3^- + {}^\bullet OH \rightarrow H_2O + CO_3^{\bullet-} \quad (k = 8.5 \times 10^6 \ M^{-1} \ s^{-1}) \ [50] \tag{15}$$

$$HCO_3^- + SO_4^{\bullet-} \rightarrow CO_3^{\bullet-} + SO_4^{2-} + H^+ \quad (k = 3.5 \times 10^6 \ M^{-1} \ s^{-1}) \ [65] \tag{16}$$

$$CO_3^{2-} + SO_4^{\bullet-} \rightarrow CO_3^{\bullet-} + SO_4^{2-} \quad (k = 4.1 \times 10^6 \ M^{-1} \ s^{-1}) \ [66] \tag{17}$$

$$Cl^- + {}^\bullet OH \rightarrow ClHO^{\bullet-} \quad (k = 4.3 \times 10^9 \ M^{-1} \ s^{-1}) \ [50] \tag{18}$$

$$Cl^- + SO_4^{\bullet-} \rightarrow Cl^\bullet + SO_4^{2-} \quad (k = 2.6 \times 10^8 \ M^{-1} \ s^{-1}) \ [66] \tag{19}$$

$${}^\bullet OH + NOM \rightarrow Products \quad (k = 2.23 \times 10^8 \ L \ (mol \ C)^{-1} \ s^{-1}) \ [67] \tag{20}$$

$$SO_4^{\bullet-} + NOM \rightarrow Products \quad (k > 6 \times 10^6 \ L \ (mol \ C)^{-1} \ s^{-1}) \ [68] \tag{21}$$

## 2.6. Performance Sustainability of the Synthesized TiO$_2$ Film

The stability and performance sustainability of the nanocrystalline photocatalyst was tested using the same TiO$_2$ film for four repeated runs. The percent degradation results achieved by the SSLA-TiO$_2$/S$_2$O$_8^{2-}$/H$_2$O$_2$ photocatalysis during four successive runs equaled a 99%, 96%, 95%, and 93% lindane removal in 6 h (Figure 7). The obtained results might indicate a high performance sustainability for the synthesized TiO$_2$ film under the experimental condition in this study. Han et al. [56] and Pelaez et al. [71] reported a similar performance sustainability using nanocrystalline S-TiO$_2$, and nitrogen and fluorine doped TiO$_2$ (NF-TiO$_2$) photocatalysts, respectively, synthesized via a similar method. Hung et al. [72] recently reported that the sol–gel synthesized doped-TiO$_2$ exhibited a higher reusability than the film synthesized by a hydrothermal method, attributable to the high calcinations temperature in the former case. The high activity and reusability may recommend the synthesized TiO$_2$ photocatalyst as a promising choice for application purposes.

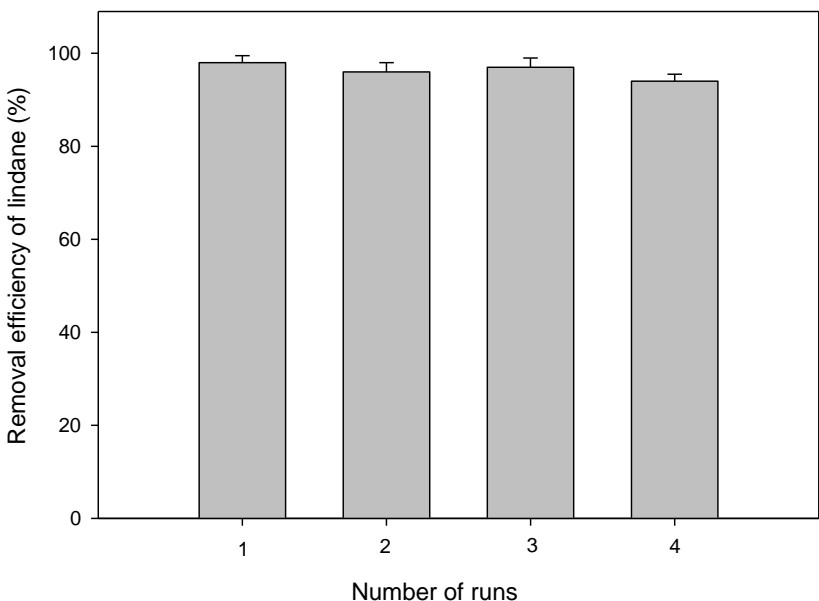

**Figure 7.** Test of reusability for the simulated solar light-activated TiO$_2$/S$_2$O$_8^{2-}$/H$_2$O$_2$ photocatalyst of lindane. [lindane]$_0$ = 1.0 μM; mass of TiO$_2$ film = 9.02 mg; thickness of TiO$_2$ film = 1.02 μm; area of TiO$_2$ film = 3750 mm$^2$; pH = 5.8.

## 3. Materials and Methods

### 3.1. Materials

Titanium (IV) isopropoxide (TTIP; 97%), lindane (C$_6$H$_6$Cl$_6$; 97%), and persulfate (S$_2$O$_8^{2-}$) were obtained from Sigma-Aldrich (St. Louis, MO, USA). Acetic acid (CH$_3$COOH, HAc), hydrogen peroxide (H$_2$O$_2$, 50%, *v/v*), sodium carbonate (Na$_2$CO$_3$), sodium chloride (NaCl), sodium nitrate (NaNO$_3$), sodium bicarbonate (NaHCO$_3$), isopropyl alcohol (i-PrOH; Certified ACS), and sodium sulfate (Na$_2$SO$_4$) were purchased from Fisher Scientific (Hampton, NH, United States). For a reference natural organic matter (NOM) in this study, standard Suwannee River humic acid (SRHA) was obtained from the International Humic Substances Society (IHSS, University of Minnesota, St. Paul, MN, USA). Benzoquinone (C$_6$H$_4$O$_2$, BQ), iso-propanol ((CH$_3$)$_2$CHOH), and tert-butanol (CH$_3$)$_3$COH) were used as radical scavengers and were obtained from ACROS Organics (Morris, NJ, USA). Milli-Q grade water (resistivity: 18.2 MΩ cm) was used to prepare the aqueous solutions during the experiments. All of the chemicals were used without further treatment.

### 3.2. TiO$_2$ Film Preparation

The preparation of film types of TiO$_2$ photocatalyst was performed with a dip-coating technique using a TiO$_2$ solution, and according to the procedure described in our previous paper [56]. In short, the TTIP was dissolved in *i*-PrOH, and then acetic acid was added in the solution. The solution was kept under vigorous stirring for 24 h at room temperature. As a result, a stable solution was obtained with a white color. The molar ratio of *i*-PrOH:TTIP:HAc was 45:1:1. The synthesized TiO$_2$ material was immobilized on a glass substrate (Gold Seal® Micro Slides (75 × 25mm; 1mm thick), Portsmouth, NH, USA) in a five-layered nanocrystalline thin film employing a dip-coating method. After the coating process, the film was dried under infrared illumination for 20 min, and then calcined at 350 °C for 2 h in a Paragon HT-22-D furnace (Thermcraft Inc., Winston-Salem, NC, USA). The temperature of 350 °C was chosen so as to avoid the formation of cracks as a result of the high stress towards the coating during the calcination process. Also, at the relatively low calcination temperature, the phase transformation from anatase to rutile, as well as the collapse of porous structure of the film were inhibited during the film preparation. The thickness of the prepared film was 1.02 ± 0.02 μm and the total mass of the immobilized TiO$_2$ was 4.51 ± 0.18 mg.

### 3.3. Characterization of Synthesized TiO$_2$ Films

The characterization techniques used in this study were similar to those in our previously published paper [56]. Briefly, the surface morphology of the synthesized TiO$_2$ photocatalyst film was characterized using an ESEM (Philips XL 30ESEM-FEG, Eindhoven, The Netherlands). The crystal size and crystal structure of the TiO$_2$ photocatalyst were characterized using a JEM-2010F high resolution-TEM (HR-TEM, JEOL, Tokyo, Japan) with a field emission gun at 200 kV. An X-ray diffraction (XRD) analysis was performed using a X'Pert PRO XRD diffractometer (Philips, Almelo, The Netherlands) with Cu Ka radiation (λ = 1.5406 Å) so as to examine the crystal structures of TiO$_2$ photocatalyst. The light absorption property of synthesized TiO$_2$ film was investigated using a Shimadzu 2501 PC UV-visible spectrophotometer equipped with an ISR 1200 integrated sphere attachment. The reference material for the analysis was BaSO$_4$. The Brunauer, Emmett, and Teller (BET) surface area of the sample was measured with a Micromeritics Tristar 3000 (Norcross, GA, USA). For the XRD, BET, UV-visible light absorption, and HR-TEM analyses, powders from the TiO$_2$ coatings were collected using a blade, which were then used for analysis.

### 3.4. Photocatalytic Experiments

The photocatalytic experiments were performed in a batch mode photoreactor of a borosilicate glass Petri dish with a diameter of 10 cm. For the UV transmission, a quartz cover was used. A 20 mL aqueous solution of lindane (1 μM) at pH 5.8 and containing two thin film-coated TiO$_2$ slides was irradiated with simulated solar light in the photoreactor. In the experiments concerning the effects of oxidants or natural water constituents on lindane degradation, desired amounts of the chemicals (i.e., 100 μM S$_2$O$_8$$^{2-}$ or H$_2$O$_2$, 1 mM inorganic ions, and 1 mg/L HA) were added to the lindane solution at the start of the experiment. A 300 W Xenon lamp (Newport, Oriel Instrument, Irvine, CA, USA) emitted simulated solar light radiation. The wavelength of the solar light was mainly from 330 to 760 nm. A schematic diagram of the photocatalytic experimentation is shown in Figure 8. The measured light irradiance ($E_e$) using a Newport broadband radiant power meter was $4.71 \times 10^{-2}$ W cm$^{-2}$. The experiments were repeated at least three times.

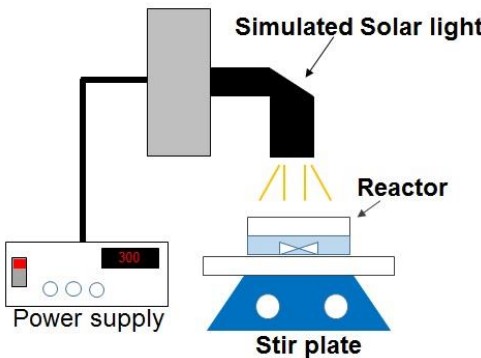

**Figure 8.** A schematic diagram of the photocatalytic experimental procedure.

*3.5. Analytical Methods*

The lindane concentration was monitored using an Agilent 6890 gas chromatograph (GC; Wilmington, DE, USA) with an Agilent 5975 mass spectrometer (MS; Wilmington, DE, USA), using the method previously reported in our study [64]. Briefly, a solid phase micro extraction (SPME) technique was used for the sample extraction. An HP-5MS (5% phenyl methylsiloxane) capillary column (length: 30 m and i.d.: 0.25 μm) was employed for the separation of the analyte. The mass spectra of the compounds in the samples were obtained in an electron impact ionization mode (EI$^+$) at 70 eV, with an *m*/z ranging from 50 to 550. An online mass spectral search program (National Institute of Standards and Technology (NIST), Gaithersburg, MD, USA, https://www.nist.gov/) was used to interpret the obtained mass spectra using the GC-MS. For the residual analysis of $H_2O_2$ and $S_2O_8^{2-}$, two colorimetric methods, described by Liang et al. [73] and Allen et al. [74], were used.

## 4. Conclusions

The efficiency of a sustainable technology employing solar photocatalysis for the elimination of lindane, a persistent organochlorine compound, from an aqueous solution was evaluated using a sol-gel synthesized nanocrystalline $TiO_2$ photocatalyst film. The results of the ESEM and TEM analyses showed a highly uniform and smooth surface morphology comprising anatase as the dominant crystalline phase in the $TiO_2$ film. The synthesized $TiO_2$ photocatalyst decomposed 23% lindane in 6 h, reflecting a limited lower removal efficiency in a practical application. Meanwhile, the efficiency of the SSLA-$TiO_2$ photocatalysis was remarkably improved by the addition of environmentally friendly and economically attractive oxidants (i.e., $S_2O_8^{2-}$ or $H_2O_2$), achieving an 89% and 64% removal of lindane, respectively, in 6 h. The removal efficiency of lindane further increased by 10% when using an equimolar mixture of both of the oxidants simultaneously (i.e., $S_2O_8^{2-}$ and $H_2O_2$), which corresponds to a 99% removal within 6 h. The $SO_4^{\bullet-}$ and $^\bullet OH$ were mostly involved in lindane degradation using an SSLA-$TiO_2$/$S_2O_8^{2-}$/$H_2O_2$ system. The lindane removal efficiency for the SSLA-$TiO_2$/$S_2O_8^{2-}$/$H_2O_2$ process decreased significantly in the presence of natural water constituents (i.e., HA, $CO_3^{2-}$, $HCO_3^-$, $NO_3^-$, $SO_4^{2-}$, and $Cl^-$), requiring a high energy input in the contaminated field water treatment processes. The sustained catalytic activity of the $TiO_2$ film during four runs manifested a high stability and reusability of the synthesized photocatalyst. The SSLA-$TiO_2$/$S_2O_8^{2-}$/$H_2O_2$ is very effective for the elimination of lindane, and potentially other OCs, from a water environment. Importantly, as the remaining $S_2O_8^{2-}$ in the process after the complete removal of OCs may result in secondary contamination [75,76], the $S_2O_8^{2-}$ concentration must be kept as low as possible in order to treat the OCs in the process. It may be the limitations of the SSLA-$TiO_2$/$S_2O_8^{2-}$/$H_2O_2$ system for practical applications.

**Supplementary Materials:** The following are available online at http://www.mdpi.com/2073-4344/9/5/425/s1. Figure S1: Tauc plot of Kubelka–Munk transformation of synthesized $TiO_2$ films. Inserts shows the UV-visible absorption spectrum of the $TiO_2$ film.

**Author Contributions:** Planning and designing of the research was made by S.K. and C.H. The experiments were performed by S.K. Materials were synthesized and characterized by S.K. and C.H. Drafting of manuscript

was done by S.K., C.H., M.S., S.J. and S.S. Critical revision was performed by D.D.D. and H.M.K. Planning and supervision of the research was made by Dionysiou D.D.D.

**Funding:** This research was funded by Women University, Swabi, Pakistan. The APC was funded by the Cyprus Research Promotion Foundation through Desmi 2009–2010, which is co-funded by the Republic of Cyprus and ERDF under contract number NEA IPODOMI/STRATH/0308/09.

**Acknowledgments:** The authors are thankful to Dominic L. Boccelli of the University of Cincinnati (UC) for allowing us to use the GC-MS. We acknowledge the financial assistance from the Women University, Swabi, Pakistan. We also acknowledge the financial support from the Cyprus Research Promotion Foundation through Desmi 2009–2010, which is co-funded by the Republic of Cyprus and ERDF under contract number NEA IPODOMI/STRATH/0308/09. Also, D.D. Dionysiou acknowledges support from the UC through a UNESCO co-Chair Professor position on "Water Access and Sustainability", and the Herman Schneider Professorship in the College of Engineering and Applied Sciences.

**Conflicts of Interest:** The authors declare no conflict of interest.

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
