# Peer review of "Exhaustive Photocatalytic Lindane Degradation by Combined Simulated Solar Light-Activated Nanocrystalline TiO2 and Inorganic Oxidants"

_catalysts, doi:10.3390/catal9050425_

Round 1
Reviewer 1 Report
The subject of the manuscript, related to solar photocatalysis, is up-to-date. The manuscript presents degradation of a model organochlorine compound, lindane, via the TiO2 photocatalysis under simulated solar light irradiation, enhanced by H2O2 and S2O82- action. The manuscript can be published after some corrections and improvements, namely:
1. Section 2.2. TiO2 film Preparation: please provide more details on glass substrate used (type of glass, dimensions, thickness, etc.). Please also explain why the temperature of 350 °C was selected for calcination.
2. Section 2.3. Characterization of synthesized TiO2 films: Please provide lindane, HA and inorganic ions concentrations. Please provide concentrations of oxidants. Please explain what you mean by “containing two films of TiO2”. The experimental procedure applied during photocatalytic tests is unclear. Please describe the experiments realized in the presence of HA, inorganic ions and scavengers.
3. Section 3.1. Characteristics of synthesized TiO2 films: this section needs thorough revision in the part related to optical properties of TiO2 films, namely:
- Eq. (2) does not represent Tauc (not Tau) equation. The Tauc equation is usually written as:
(αhν)^(1/n)=A(hν-Eg),
whereas to calculate the Eg value, a plot of (αhν)^(1/n) = f(hν) is applied.
Moreover, (hν) represents photon energy, not “photon energy of the synthesized TiO2 photocatalyst”. For indirect semiconductors, n =2.
- “Figure 1 (e) shows that the commercial TiO2 and the synthesized TiO2 have a broad 161 absorption band in the UV-Visible region (350–600 nm) (Figure 1 (e) inset).” – from this Figure one may observe that commercial TiO2 does not absorb in the visible range, it absorbs below 400 nm only
- Please explain what you mean by “for the TiO2 photocatalyst with the synthesis method” (line 155) – this expression is unclear
- Lines 165-167: “This suggests that the adopted synthesis procedure decreased the Eg value of the synthesized photocatalyst, which could be due to reduction in the particle size of TiO2.” – this should be discussed in more details with reference to the quantum size effect
- Lines 167-168: “free electrons are excited in the valence band” – it should be “electrons are excited from the valence band to conduction band”; Please delete the sentence “The excited free electrons move from the valence band into the conduction band of TiO2.” because it repeats the above information. Furthermore, please use “electrons”, not “free electrons’;
- Lines 169-171: “The free electrons in the conduction band interact with oxygen and water molecules adsorbed on the surface of the TiO2 film, which lead to the formation of reactive oxygen species [24].” – in fact electrons react with O2, while holes react with H2O or OH- - please rewrite this sentence
- Figure 1: it is not clear how the spectra were collected: was the commercial TiO2 deposited on the glass substrate, or the powdered sample was measured? How the synthesized sample was prepared for the measurement? Please provide more experimental details in Section 2.
4. Section 3.1.: How was the BET surface area of the synthesized sample measured? Was the TiO2 film deposited on the glass plate analyzed? If so, what was the BET surface area of the glass plate? More details should be provided in Section 2.
5. Section 3.2.: equation (3) is not correct (it is not balanced) – please write two separate equations for OH- and H2O
6. Captions of Figures 2-7: giving the TiO2 concentration is misleading – it suggests that TiO2 suspension, not TiO2 deposited on a support was used. Moreover, in case of deposited TiO2 only a part of the particles area is available for the reaction. Therefore, it would be better to provide here TiO2 mass and film thickness and area instead of concentration.
7. Section 3.2. Lines 205-212: This part of discussion is not necessary: it is obvious that generation of OH* radicals is higher under UV than solar radiation. Thus, the paragraph should be shortened giving just the conclusion or deleted.
8. Section 3.3. Lines 248-256: this part of discussion is not confirmed by the results shown in Fig. 3. Fig. 3 presents changes in lindane concentration only, and does not represent any by-products of degradation or mineralization efficiency. Please rewrite this paragraph or support this part of discussion by relevant data (e.g. total organic carbon measurement).
9. Section 3.3. Figure 3 caption: please provide concentrations of oxidants in the SSLA-TiO2/S2O82-/H2O2 experiment
10. Section 3.4. equations (11)-(13) – please write them in separate lines
11. Section 3.4. Figure 5 caption: please provide scavengers concentrations
12. Section 3.5. Lines 307-310: please combine “The reactive radicals can also react with the inorganic ions [17]”and “as well as scavenging of the reactive radicals by the ions [55, 58].”in one sentence.
13. Section 3.5: please explain why so low HA concentration (1 mg/L) was applied.
14. General comment for the discussion: it would be interesting to show concentrations of H2O2 and SO42- after the treatment process: were the values monitored?
15. Conclusions should be reorganized. Please avoid repetitions (e.g. line 369 vs. lines 379-380). Moreover, the mechanical strength (line 376) was not investigated, thus this conclusion is not supported by the data.
16. English needs revision – there are some minor spelling and language mistakes, e.g. line 7: Departmen, instead of Department; line 32: presence water constituents, instead of presence of water constituents; line 158: reaction instead of equation; line 180: Figure 2 shows that the… instead of Figure 2 shows the…; line 219; Figure 3 shows that the… instead of Figure 3 shows the…; line 316: in t the instead of in the; Line 328: please rewrite the sentence (e.g.: The efficiency of lindane degradation by SSLA-TiO2/S2O82−/H2O2 process decreased by 39%), etc.
17. All abbreviations should be explained when used for the first time (including abstract), e.g. HA (line34), DOMs (line 334), S-TiO2 and NF-TiO2 (line 352), etc.
Author Response
Uploaded as a Word file.

Reviewer 2 Report
The manuscript by Khan and co-authors is an interesting piece of work dealing wit the degradation of a dangerous organochlorine compounds.
The paper is well written, conclusions are well supported by results.
The only comment I feel to do is at page 4, lines 166-167.
Authors justify that the low (2.8 eV) Eg of their material is due to "reduction of TiO2 particle size". In other words, to quantum size effects.
However, this entails that particles becomes smaller than their Bohr radius. This is reported to be around 2.5 nm for anatase (see for instance: DOI: 10.1016/j.jphotochemrev.2011.02.001). According to the reported TEM images, this doesn't seem the case.
Please justify that behaviour in a more convincing way.
Author Response
Uploaded as a Word file.

Reviewer 3 Report
Please find the attachment.

Author Response
Uploaded as a Word file.

Round 2
Reviewer 1 Report
Authors have addressed all my comments and remarks. The manuscript can be accepted after a minor correction:
In figures captions the mass of catalysts is given in wrong units (g/L) instead of (g). Please also check if the value (i.e. 4.6) is correct.
Reviewer 3 Report
The quality of revised manuscript improved after 1st revision. However the below comments should be taken care of -
It is strongly recommended that the band-gap energy of the synthesized TiO2 film should be mentioned. Authors should Include the UV-Vis spectrum and characterization details of the material. This is important in regard to the photo-catalytic application of the material.
What is wavelength of light used in the photo experiment?
Comment 4. L-164: “…results showed that the Eg of commercial and synthesized TiO2 was observed to be 3.2 and 2.8 eV, respectively. Authors mentioned the Band gap energy, Eg of their synthesized TiO2 is ~ 2.8 eV. But they claimed that synthesized TiO2 is predominantly anatase. Most of the research article shows (see above refs.) that Anatase has higher Eg ~3.2-3.0 eV, while rutile has relatively lower Eg ~2.8 eV. Can the authors explain their findings clearly?
Answer: Thank you very much for your comment on this discussion. We accidentally provided wrong information. We corrected the part in the revised manuscript.
Reviewer: Authors need to provide the band gap energy and justify the comment#4.
Please correct these as well-
Meanwhile, the efficiency of 500 SSAL-TiO2 photocatalysis was...- typo 'SSAL'
The temperature of 350 oC was chosen to avoid the formation of crack due to high stress towards the coating during the calcination process. "- Is this only reason to choose calcination temp. 350 DegC? Should not be associated with preserving anatase morphology and crystallite size, band gap energy etc. as change in temp. would have modified these properties and subsequently photocatalytic performance.
